# Association between the Mother’s Social Cognition and the Child’s Social Functioning in Kindergarten: The Mediating Role of the Child’s Social Cognition

**DOI:** 10.3390/ijerph17010358

**Published:** 2020-01-05

**Authors:** Yair Ziv, Reout Arbel

**Affiliations:** Department of Counseling and Human Development, University of Haifa, Abba Khoushy Ave 199, Haifa 3498838, Isreal; reout.arbel@gmail.com

**Keywords:** social cognition, social information processing, mother-child relationships, parenting style, kindergarten, social functioning

## Abstract

Children’s ability to adjust to the social rules and expectations in the educational environment is of major concern to researchers and practitioners alike. Accordingly, the main purpose of the present study was to examine predictors of children’s social functioning in kindergarten with a specific focus on (a) maternal factors and (b) children’s social cognition. Using a multi-method (self-reports and direct assessments), multi-informant (child, mother, teacher) design, we collected data from 301 kindergarten children and their mothers tapping the mother’s social cognition (general and child-related) and parenting style, and children’s social cognition (social information processing) and functioning in kindergarten. We found direct associations between the mother and child’s social cognition, between the mother’s authoritarian parenting style and her child’s less competent social cognition and behavior, and between the child’s social cognition and social functioning. Finally, as hypothesized, we found a number of interesting mediated effects. Most notably, we found that the association between the mother’s social cognition (her tendency to attribute hostile intent to unknown others) and the child’s social cognition (his/her tendency to generate less competent responses) is fully mediated by the mother’s higher levels of authoritarian parenting style. The important theoretical and clinical implications of our findings are discussed.

## 1. Introduction

Children’s social cognition, their ability to understand and think about others’ mental states in social situations, plays an imperative role in the development of their social relationships (e.g., [1,2,3,4]). Higher social cognitive capacities are typically associated with prosocial skills whereas difficulties in understanding and thinking about others’ mental states can lead to disruptive social functioning and increase the risk of major mental health problems [5]. These social cognitive processes are shaped in the context of close relationships across child development and, in particular, within the context of the child’s relationship with his/her main caregivers [5,6,7,8,9,10]. Accordingly, the main purpose of the current study is to examine the associations between mothers’ parenting characteristics and general social cognition and their children’s social perceptions and behaviors. More specifically, we are interested in the potential mediating role of parenting characteristics in the link between mother and child’s social cognitive capacities (i.e., their information processing patterns), and the mediating role of the child’s social information processing in the link between the mother’s parenting characteristics and the child social functioning in kindergarten. The expectations for significant (direct and indirect) associations between these variables are firmly grounded in two major developmental models/theories: Attachment theory [11,12], which attempts to describe the dynamics of long-term interpersonal relationships between humans, particularly in the context of a parent–child relationship, and Crick and Dodge’s [13] social information processing (SIP) model, which focuses on humans’ subjective perceptions and interpretations of their social environment and the ways by which these subjective assessments guide social behavior. Our conceptual model linking parent and child social cognition and functioning is based on these two models.

### 1.1. Associations between Mother and Child Social Cognition Via Parenting Characteristics

A core assumption of attachment theory is that external behaviors are guided by internal mental representations [12,14], which are instrumental in shaping one’s state of mind. Thus, social cognition is at the heart of this theoretical model. Accordingly, the theory assumes that a caregiver is more likely to provide positive parenting when her perceptions of the surrounding social world are positive. When a mother’s state of mind and perception of her social world is characterized by flexibility and non-defensiveness, it is more likely that she can adjust her behavior based on the correct read of the child’s current state of mind. On the other hand, if the caregiver perceives the social world as threatening and hostile, this cognition is likely to be echoed in a parenting style that is less reflective and more defensive, which, in turn, will affect the child’s own social cognition. Thus, the main assumption of this theory is that parents’ own social cognition affects their children’s social cognition via the quality of parenting.

Fonagy and colleagues’ furthered attachment theory’s discussion on internal representations by introducing the important theoretical concept of “mentalization” [15]. They describe this concept as the mental ability to produce internal representations of self, others, and the self in interaction with others that are used to process information about the self and the others’ states of mind. Individuals who mentalize effectively can better regulate affect and, consequently, are more likely to better function in social situations, whether familiar or novel. Conversely, ineffective mentalization, i.e., the inability to be reflective about others’ thoughts and feelings, is likely to result in lower social skills and major functioning difficulties [15,16]. Fonagy and colleagues also discuss the consequences of ineffective mentalization on parenting. They suggest that ineffective mentalization is associated with specific early experiences that affected the parent’s ability to effectively (or correctly) interpret her child’s state of mind as the parent is too absorbed in her own, and possibly traumatic, early experiences [17,18]. On the other hand, when a parent is able to connect past experiences with internal representations, she is more likely to better understand and accept the child’s current states of mind. This will likely result in satisfactory interpersonal relationships. A parent that is characterized as reflective in the context of her relationship with her child is able to view the child through his or her own eyes, to effectively interpret situations from the child’s perspective, and to reflect on the child’s mental states based on actual observations. 

In continuing and expanding on this line of thought, Mitchell [19] suggests that, in order to better understand how humans make mental state inferences, we should treat specific social cognition primarily as part of a wider information processing system. This system includes a set of mechanisms that can translate more basic social information components into a complete perspective of the internal state of mind. More specifically, Mitchell calls for connecting more specific social cognition (in this case, for example, parental perception of the relationship with the child) to more general information processing capabilities that guides the parent’s behavior in general (i.e., their general view of the social world). Corresponding to this call, the study aims to show a connection between the mother’s general SIP patterns (i.e., the way she generally perceives the social world surrounding her) and her relationship-specific social cognition (i.e., her perception of the relationship with the child). Based on the theoretical path suggested by attachment theory, it is hypothesized that the mother’s general SIP patterns will be related to her parenting behaviors and relationship-specific social cognition (i.e., her perception of the relationship with the child), and via both, to the child’s perceptions and behaviors.

### 1.2. Associations between Quality of Parent-Child Relationship and Children’s Behavior Via SIP

Crick and Dodge [13] proposed a theoretical path by which the quality of early relationships influences children’s social functioning in various social situations through their social information processing patterns. In their formative SIP model, early relationships are viewed as the basis for a database of social knowledge that guides the enactment of a social response. When children encounter new social situations, they access knowledge available to them from previous experiences about the intentions of others in their environment, the meaning of the others’ actions, and the best ways to act upon these actions. Especially in young children, the database which guides their behavior is grounded in experiences with other family members, and in particular, with their parents. The model further suggests that mental representations affect social behaviors through on-line social information processes, the mental processes that come into action when a social input is received and before a social output is produced. These mental processes include: (1) Encoding, (2) interpretation, (3) clarification of goals, (4) response construction, and (5) response decision. The sixth step in the model is the actual enactment of a behavioral response [13]. (see Figure 1)

As mentioned, attachment theory [11,12] portrays a similar perspective on the ways by which children form expectations about their social world and act upon them. Bowlby argues that children form internal working models (IWMs) of relationships based on their experiences with their attachment figures, and that these models shape their thought processes and social functioning. He characterizes IWMs as internal mental representations that are based on real interaction experiences between individuals and their principal attachment figures. These experiences are translated into generalized mental representations of self and others that reflect the degree to which the individual feels worthy of others’ care and affection (this is the person’s generalized model of self) and the degree to which he/she perceives others as available, empathic to his/her own needs, and responsive to these needs (this is the person’s generalized model of others). Bowlby links emotionally available (or sensitive) parenting with secure attachment and insensitive parenting with insecure attachment. Secure attachment is reflected in open, flexible, and non-defensive mental representations, whereas insecure attachment is reflected in mental representations that are typically defensive, biased, and resistant to changes. The theory refers to these models as schematic processes that facilitate or limit access to information about social relationships [20], a description very similar to that of Crick and Dodge’s. Indeed, in a systematic review of SIP and attachment, Dykas and Cassidy [21] suggested that attachment theory should be looked at as a valuable framework that can facilitate a better understanding of the mechanisms by which early relationships guide generalized SIP patterns. They maintained that the caregiver, by providing (or not providing) a secure base, helps in shaping the child’s current and future SIP patterns. These, in turn, will come into action in various social situations and in the context of different social relationships. Thus, the child forms expectations for new social interactions that are largely based on his/her past experiences with the attachment figure. In that sense, these interpretations do not represent an “objective” look of the surrounding social world but are rather a reflection of the child’s internal working models of self and others.

### 1.3. Direct Association between Parenting Characteristics and Children’s Social Perceptions and Behaviors

The association between parenting styles and children’s social behavior is well-established. For example, less competent parenting styles, such as authoritarian or permissive parenting, have been frequently linked to higher levels of children’s aggressive and withdrawn behaviors in school (e.g., [22,23]), while authoritative (i.e., a parenting style combining structure and sensitivity) parenting style has been found to predict children’s prosocial behavior and, mainly, the absence of problem behavior (e.g., [24,25,26]). In a secondary-analysis study using the NICHD Study of Early Child Care longitudinal dataset [27,28], Fraley et al. [29] found that sensitive parenting assessed in infancy, as well as in later points across development, strongly predicts children’s social competence in childhood and adolescence. In another study using the same dataset, sensitive parenting in infancy predicted children’s better social problem-solving skills in school [25]. 

In a different longitudinal study with children aged 7–10 [30], the researchers examined the effects of parenting style on children’s adjustment to school. As expected, findings showed that negative parenting predicted higher social dysfunction in school, whereas positive parenting (e.g., warmth) predicted higher social skills and fewer behavior problems in school. In relation to the specific link between parenting and SIP, a number of studies highlighted especially the effect of negative parental patterns, such as negative emotionality, criticism, covert and overt hostility, and other characteristics of authoritarian parenting style, on children’s more negative social information processing in social contexts outside the family, particularly in school (e.g., [31,32,33,34]). Only four studies examined the connection between parenting behaviors and children’s SIP patterns, as well as children’s behavior in preschool or school [35,36,37,38]. Using the NICHD ECCRN dataset, McElwain et al. reported on a link between mother–child affective mutuality in kindergarten and fewer hostile attributions and greater peer competence in first grade [35]. On the other hand, Runions and Keating [37], using the same dataset, reported that negative parental control did not predict SIP at all [37]. In the two other studies, Ziv and colleagues [38] reported on significant links between negative maternal control and children’s less competent selection and evaluation of responses, and Pettit and colleagues [36] reported that parent intrusiveness was positively related to children’s tendencies to select aggressive responses whereas positive parental involvement was related to children’s tendencies to select competent responses.

### 1.4. Direct Association between Children’s Social Information Processing and Their Social Functioning

Associations between social information processing and social behavior have been reported in numerous studies. Most of these studies have reported significant associations between negatively biased SIP patterns and maladjusted behavior in school. For example, aggressive children tended to ascribe hostile intents to peers in benign situations (e.g., [39,40,41,42] and to propose aggressive solutions to such situations [42,43,44]. These children also thought that aggressive behavior is likely to produce positive instrumental (i.e., the other children will let them play) and interpersonal (i.e., the other children will like them) outcomes for aggressive responses [42,45]. Similar, yet more refined, results were found for withdrawn and shy children. For example, Burgess and colleagues found that these children are more likely to attribute hostile intents to peers, but only if the peers are unfamiliar [46]. Likewise, children characterized as victims were reported to avoid challenging social situations altogether because of initial expectations that others are purposefully hostile or ignoring [1,47]. In contrast, prosocial children have been found to exhibit highly competent SIP patterns in all stages of the process (e.g., [48,49]). 

Similar trends were found in the small number of studies examining the associations between SIP and social behavior in preschool and kindergarten children. Katsurada and Sguwara [50] showed that preschoolers characterized as aggressive are more likely than other children to attribute hostile intent to others. Importantly, these researchers also found that already in preschool, children are capable of distinguishing between intentional and unintentional actions. In other studies with preschool children, Hart and his colleagues [51] reported that children exhibiting social difficulties in preschool anticipate more positive instrumental outcomes for aggressive resolutions of conflict, and Runions and Keating [37] showed that hostile attribution in preschool is a better predictor of problem behavior in first grade than hostile attribution measured concurrently (also in first grade). In more recent studies, Denham and colleagues reported on associations between preschoolers’ SIP patterns and social adjustment in elementary school [52], and Schultz and colleagues reported similar findings with a specific focus on two SIP stages: Response construction and response decision [53]. Finally, Ziv and colleagues showed that negatively biased SIP patterns in preschool predict problem behaviors in kindergarten, and, in addition, are negatively related to positive social skills [1,38,54,55].

### 1.5. Hypotheses

Based on the above review, we hypothesize that a complex set of associations exists between the mother’s perceptions (general SIP and perception of relationships with the child) and behavior (parenting style) and the child’s perceptions (SIP) and behaviors (social functioning). These associations are portrayed in Figure 2 (our conceptual model) and are summarized in the following two main hypotheses: (1)The mother’s general SIP patterns will be significantly associated with her child’s SIP patterns, but this association is expected to be mediated by (a) the mother’s parenting style, and, separately, (b) by the mother’s perception of relationship with the child.(2)The mother’s perceptions of her relationship with the child will be associated with the child’s social functioning, but this association will be mediated by the child’s SIP patterns.

## 2. Materials and Methods 

### 2.1. Participants 

Three hundred and one children (152 girls, 149 boys, mean age 5.72 years, SD = 0.53) and their mothers participated in the study. Data were collected between the years 2016 and 2019 in a large metropolitan area in the north part of Israel. About 80 mothers were college-educated, and all were married. Mothers had, on average, 2.83 (SD = 1.11) children. Family income was rated on a five-point scale. The question about income included the average monthly income in Israel per family (about $4000 as of 2014) and parents were asked whether their income was much below this mean (1), below the mean (2), about equivalent to the mean (3), above the mean (4), or much above the mean (5). Thus, the mean score in the study of 3.54 suggests that the mean family income in this sample was slightly above the country’s average. More information about the sample could be found at the bottom part of Table 1.

### 2.2. Research Measures

Measures are reported as they correspond to the study’s main theoretical constructs: Maternal constructs (predictors: General SIP, parenting style, perception of the child), child’s SIP patterns (mediator/outcome) and child’s social functioning in kindergarten (outcome).

#### 2.2.1. Maternal Constructs

**Mothers’ general SIP** patterns were assessed using a questionnaire that is based on the Social information processing-attribution and emotional response [56]. In this questionnaire, the mother is asked to respond to a series of hypothetical scenarios in which other individuals are behaving in ways that could be interpreted as intrusive, rude, and anti-social, however, this interpretation is interrupted by what could be conceived as a strong rationale that justifies this behavior in the particular situation. For example, in one of the scenarios, a person is jumping a line in the supermarket in front of the protagonist but says she is doing it because she is in a hurry to catch a bus (thus, jumping a line could be perceived as anti-social, but the person is providing a rationale that could be conceivable). In total, four different scenarios are used. After each scenario, mothers were asked multiple-choice questions about the intents of the other person (e.g., is she lying or not), what she (the mother) would do in a situation like this (e.g., will she let the other person pass), will she say anything to that person, and what she would do in that same situation (e.g., what she would do if she was in the supermarket in a big hurry when there is a big line). Additionally, mothers were asked a series of Likert-like questions about intents and actions in similar situations and asked how much they identify with a specific action/thought (five points scale: From highly identify to highly do not identify, e.g., “I will let the woman pass and act nicely to her, I will ignore the woman and will not let her pass). In the current study, one summary score, mother’s hostile attribution bias, was used. The possible range of this score was 0–7, with higher scores representing higher levels of hostile attribution bias.

**Mother’s parenting style** was measured using the Parenting Styles and Dimensions Questionnaire (PSDQ; [57]). The 32-items questionnaire identifies three parenting styles: Authoritative (15 items), authoritarian (12-items), and permissive (five items). An example of an authoritative item is: “I listen to my child when I need to make a decision, but I do not make a certain decision just because my child wants me to”. An example of an item targeting authoritarian style is: “I slap my child when s/he misbehaves”. Finally, an example of an item targeting permissive behavior is: “I allow my child to make his/her own decisions, without a lot of help from me”. The questionnaire was reported to have good psychometric properties, including in Israel (e.g., [58]). In the current study, Alpha reliability scores for the authoritative and permissive scales were very low (lower than 0.50 for both scales) and thus these two scales were excluded from further analyses. Alpha reliability scale for the authoritarian scale was 0.83 and thus we used this scale as the only marker of parenting style in this study.

**Mother’s perception of the relationships with the child** was measured using the short-form child–parent relationship scale (CPRS; [59]). The scale includes two different scales: Conflict (e.g., “my child and I always seem to be struggling with each other”), and closeness (e.g., “I share an affectionate, warm relationship with my child”). Each of the 15 items is scored on a five-point Likert scale (from 0—definitely does not apply, to 4—definitely applies). Reliability score (Alpha) for the conflict and closeness scales were 0.72 and 0.65, respectively.

#### 2.2.2. Child Constructs

**Social information processing patterns** were measured using the social information processing interview, preschool version (SIPI-P; [55]). This 20-min structured interview is based on a storybook easel depicting a series of four vignettes in which a protagonist is either being excluded by two peers (the two peer-exclusion vignettes) or provoked by another peer (the two peer-provocation vignettes). The peers’ intent is portrayed as either ambiguous or non-hostile/accidental (never intentionally hostile). The illustrations in the storybook are of cartoon bear characters and there are parallel picture books for boys and girls. As the child hears the story, the interviewer stops at scripted points and poses questions addressing the hypothesized information processing steps. Eight main scores are initially derived from the SIPI-P: (1) Efficient encoding (α = 0.84), which is a summary score of the child responses to the question (asked once for each of four stories): “What happened in the story, from the beginning to the end” with higher scores representing better recollection, (2) hostile attribution bias (α = 0.69), which is a frequency count of the number of times the child describes the other child/ren as having hostile intents across the four stories (based on the question: “Were the other child/ren mean or not mean?”). Thus, the range for this score is 0 to 4 with higher scores representing higher tendency to attribute hostile intent to peers, (3) competent response generation, (4) aggressive response generation, and (5) inept response generation. Each of these three scores represents a summary of the child’s responses to the question: “What would you do if this (whatever happened in the said story) happened to you?” The possible range of each of these scores is 0–4 with higher scores representing higher levels of competent/aggressive/inept response construction, respectively, (6) competent response evaluation (α = 0.87), (7) aggressive response evaluation (α = 0.80), and (8) inept response evaluation (α = 0.86). Each of these three scores represents a summary of the children’s evaluation of a response (i.e., competent, aggressive, or inept) presented to them (e.g., the child is shown an aggressive response, for example, the child ruins the other children’s game, and is asked three questions: “Was this a good thing or a bad thing to do?”, “if you had done this, will the other children love you?”, “if you had done this, will the other children let you play?”). The possible range of each of these scores is 0–12 with higher scores representing higher levels of competent/aggressive/inept response evaluation, respectively (for more information about this measure see [55]).

**Children’s social functioning** was measured using the Strengths and Difficulties Questionnaire, teacher version (SDQ; [60]). The SDQ is a short behavioral screening questionnaire that is designed to assess the behaviors of children ages 4–16 years old focusing on five main attributes: Emotional symptoms (e.g., often complains of headaches, stomach-aches, or sickness), conduct problems (e.g., often fights with other children or bullies them), hyperactivity/inattention (e.g., restless, overactive, cannot stay still for long), peer relationship problems (e.g., picked on or bullied by other children), and prosocial behavior (e.g., considerate of other children’s feelings). Each of the five scales includes five items for a total of 25 items, all rated on a three-point Likert scale: “Not true” (scored ‘0′), “somewhat true” (scored ‘1′), and “certainly true” (scored ‘2′). Thus, the possible score range for each of the five scales is 0–10 (with the possibility of combining the four “problems” scales into one 0–40 “problem behaviors” score). The questionnaire was reported to have adequate reliability and validity, with Cronbach α for all scales ranging from 0.69 to 0.84 (e.g., [61,62]). In the current study, Alpha reliability scores ranged from 0.52 (peer problems) to 0.81 (hyperactivity and prosocial scales) for the individual scales. The alpha for the combined negative scale (combining the four negative scales) was 0.84. 

### 2.3. Procedure

After receiving approval for the study’s protocol for the University’s IRB (approval # 464/16) as well as from the Department of Education chief scientist office (approval # 9312), we contacted the families through fliers distributed in the kindergarten classrooms. After receiving signed consents from parents, we contacted the families to schedule a home visit in which parents completed questionnaires assessing their general SIP patterns, their perceptions of the relationships with the child, and their parenting styles. Additionally, they provided demographic information, such as parental level of education, family income, number of children, and other family members in the household, etc. Next, we scheduled classroom visits in which we directly assessed the child’s SIP patterns. Finally, the child’s teacher completed a questionnaire reporting on the child’s social functioning in the classroom. Thus, information collected in this study was obtained from three separate reporters (mother, child, and teacher), on three separate occasions.

### 2.4. Analytic Strategy

To test our hypotheses, we conducted path analysis in Mplus Version 7 [63]. Indirect effects were tested with the bootstrap method with 95% confidence intervals. All models ran as saturated. Mediation effect sizes were calculated as the relative magnitude of the indirect to the total effect. We tested the potential contribution of background covariates that associated significantly with the specific outcome (SIP or social functioning). Specifically, in the prediction of child’s SIP patterns, ethnicity (sector) and mother’s education were tested. In the prediction of child’s social functioning, child’s sex and the mother’s education levels were tested. If including those covariates did not change the patterns of significance or magnitude of effects, they were excluded from the model for parsimony [64].

Our model includes five different levels of data: (a) The mother’s general SIP patterns, (b) the mother’s parenting style, (c) the mother’s perception of the relationship with the child, (d) the child’s SIP, and (e) the child’s social functioning. Because each of these levels included multiple observed variables, we ended up using the variables from each level that corresponded best with our hypotheses and showed good reliability (see measures section above). We ran a number of models with different predictors and outcomes and we present here the models that showed the highest promise. Accordingly, our final operational model include the following five variables: Level 1 = Mother’s hostile attribution bias (M-HAB), level 2 = Mother’s authoritarian parenting style (M-Authoritarian), level 3 = Mother’s perception of her relationship with the child as conflictual (M-conflict), level 4 = child incompetent response generation (a reversed competent response generation score-C-ICRG, to match the direction of the other four variables in the model, i.e., higher scores equals more negative behaviors and perceptions), and level 5 = Child’s total problem behaviors (C-PB). The conceptual model portrayed in Figure 2 is shown again in Figure 3, this time with the actual observed variables used in our main analyses. 

## 3. Results

### 3.1. Descriptive Statistics and Intercorrelations

Table 1 presents the means, standard deviations, and bivariate correlations among the study’s five main variables and a number of potential background moderators: Child’s age and sex, mother’s education, family income, and sector (Jewish or Arabs). As can be seen in the table, two background variables were particularly associated with the study’s main variables: Mother’s education and sector, and more sporadic associations were found for the other background variables. As for the associations between the study’s main variables, the mother’s hostile attribution bias was positively associated with authoritarian parenting style and with higher levels of conflict, as well as with higher levels of incompetent response generation by the children. The mother’s authoritarian style was strongly associated with the level of conflict in the relationships and positively associated with incompetent response generation and the child’s problem behavior. The mother’s conflict was positively associated with incompetent response generation and with the child’s problem behavior. Finally, the child’s incompetent response generation and total problem behavior were positively associated. Thus, all the preliminary conditions for examining the mediations in our main analyses were fulfilled.

### 3.2. Main Analysis

#### 3.2.1. Hypothesis 1: Parenting Factors Mediate the Effect of Mother’s SIP on Child’s SIP

We tested the indirect effect of the mother’s hostile attribution bias on the child’s incompetent response generation through the mother’s authoritarian parenting style and separately through her positive perception of the relationship with the child. As shown in Figure 4, authoritarian parenting style fully mediated the association between the mother’s hostile attribution bias and the child’s incompetent response generation. Specifically, the mother’s hostile attribution bias had a positive total effect on the child’s incompetent response generation (B = 0.17, SE = 0.05, *p* = 0.04, CI [0.01,0.21]) and predicted higher authoritarian parenting style (B = 0.32, SE = 0.06, *p* < 0.001, CI [0.21,0.43]). Higher authoritarian parenting style, in turn, predicted the child’s incompetent response generation (B = 0.24, SE = 0.06, CI [0.07,0.33]). The indirect effect between the mother’s hostile attribution bias and the child’s incompetent response generation through authoritarian parenting style was significant (B = 0.06, SE = 0.03, CI [0.02,0.12]). Finally, after adjusting for authoritarian parenting style, the direct effect of the mother’s hostile attribution bias on the child’s incompetent response generation was no longer significant, (B = 0.09, SE = 0.06, CI [0.04,0.21]). The indirect effect through authoritarian parenting style accounted for 35% of the total effect of the mother’s hostile attribution bias on the child’s incompetent response generation. When we controlled for child’s ethnicity or mother’s education the indirect effect remained significant, (B = 0.03, SE = 0.02, CI [0.00, 0.06]).

Next, we tested the indirect effect of the mother’s hostile attribution bias on the child’s incompetent response generation through the mother’s perception of the relationship with the child as conflictual. The indirect path was found significant (B = 0.03, SE = 0.02, CI [0.004, 0.05]), explaining about 17% in link between mother and child’s SIP. However, after we adjusted the models for child’s ethnicity and mother’s education the mediation effect became null, (B = 0.02, SE = 0.02, CI [−0.001, 0.05]).

#### 3.2.2. Hypothesis 2: Child SIP Mediates the Effect of Mother’s Parenting Factors on Child’s Functioning 

In two parallel models, we tested the mediating role of child’s SIP in the link between mother’s perception of the relationship with the child’s social functioning, and separately between mother’s parenting style and child’s social functioning. The mediation effect of SIP on the link between authoritarian parenting style and child functioning approached significance (B = 0.03, SE = 0.06, CI [−0.004,0.07]) and became null once we adjusted the models for youth sex and the mother’s education. Similar results emerged for the mediating effect of SIP on the link between conflict and child’s functioning, which approached significance, (B = 0.03, SE = 0.02, CI [−0.006, 0.01]), and became null when we adjusted for youth sex and the mother’s education.

## 4. Discussion

Children’s ability to adjust to the social rules and expectations in the educational environment is of major concern to researchers and practitioners alike. This is especially true in the preschool and kindergarten transition years when children are about to enter the much more demanding primary school system and the types of behaviors they demonstrate could strongly affect their ability to transition successfully. On the other hand, these early years are also an opportunity as the earlier specific social adjustment difficulties are detected, the chances of successful intervention increase. So that such interventions are successful, it is critical to understand the antecedents that contribute to more or less successful social functioning. Accordingly, the main purpose of the present study was to examine predictors of children’s social functioning in kindergarten with a specific focus on (a) maternal factors and (b) children’s social cognition. We posed two main hypotheses: First, we hypothesized that the association between mother and child’s general social cognition will be mediated by the mother’s parenting style and by her perception of the relationship with the child. Second, that parenting factors (style and perception of the relationship with the child) will be associated with the child’s social functioning in school through the child’s social cognition. These hypotheses were partially confirmed in this study: Authoritarian parenting style indeed fully mediated the association between mother and child’s SIP. On the other hand, the other hypothesized mediated paths became null when other background factors (sector, maternal education, and child sex) were entered into the equation. These findings have both theoretical and clinical implications that are discussed next.

### 4.1. Theoretical Implications

Our findings generally support the two theoretical paths presented at the onset of this article. In these hypothesized paths, (a) the mother’s SIP patterns affect the child’s SIP patterns through her parenting behaviors and perceptions, and (b) the quality of the parent–child relationship contributes to the explained variance in the child’s mental representations of the social world (SIP patterns) which, in turn, contributes to the explained variance in the child’s exhibited social functioning in school. Further, our findings highlight the specific components of these hypothesized paths, which in our sample, like in most previous studies on these links, are more strongly related to negative behaviors and perceptions of the mother and child rather than to positive ones. Notably, these links were found while using data from multiple independent sources (mother, child, and teacher) and with different measurement approaches (questionnaires and direct assessments). This multi-method, multi-informant approach accentuates the validity and power of its findings.

The strong indirect link between the mother and child’s SIP through parenting style is especially notable. To remind the reader, information about the mother and child’s SIP was obtained by asking the members of the dyad questions that are not directly related to the other member. Mothers were asked about their interpretations of hypothetical common scenarios in the adult world (e.g., someone they do not know is trying to skip a line), and children were asked to respond about hypothetical scenes in the kindergarten environment (e.g., other children do not let the protagonist child join them in a game). Thus, the direct link found between the two is not trivial. It suggests that by some means, the general perception of the parent about the social world is transferred to the child so that the child’s view of his/her social world resembles the parent’s. The current study provides indications about the means by which this intergenerational transference occurs. Most notably, it seems that the mother’s general perception of the social world affects her parenting style. This seems to be especially noticeable in the case of authoritarian parenting style. In our study, mothers who viewed the social world more suspiciously (i.e., attributed more hostile intents to unknown individuals), also seemed more comfortable reporting on being harsher with their child (and therefore receiving higher scores on the authoritarian scale). This, in turn, affected the child’s perception of his/her own social world as children with authoritarian mothers were less inclined to generate competent social solutions to complex and challenging social situations. The mediation analysis revealed that the previously significant association between the mother and child’s SIP is no longer significant when authoritarian parenting style is entered into the equation and that more than a third of the link between mother and child’s SIP is explained by this parenting style. Thus, one means by which the parent’s general perception of her social world is transferred to the child is through the mother’s behavior toward the child and this transference is apparent only in the case of negative perceptions of the world (by both members of the dyad).

Second, the same link (mother to child’s SIP) was also mediated by the way the mother perceived her relationship with the child. Mothers who tended to attribute more hostile intents to unknown others also tended to view their relationships with their children as more conflictual than mothers who did not have the same tendency. In turn, their children’s SIP patterns were less competent. Although the indirect effect, in that case, did not hold after entering background factors into the equation, the mediation analysis still revealed that a significant portion of the link between mother and child’s SIP is explained by the mother’s perception of her relationship with the child thus providing evidence for another means by which the parent’s general perception of her social world is transferred to the child, the mother’s perception of her relationship with the child as conflictual.

Importantly, authoritarian parenting style was also strongly related to the mother’s perception of the relationship as conflictual. Mothers who reported to behave harsher with their children also reported more conflict in the relationship. Whereas this association is important on its own merit, it also suggests that an alternative path to the one suggested in our theoretical model is possible, i.e., that the high level of conflict leads the parent to behave more harshly towards the child and not the other way around: That authoritarian parenting style leads to more conflict in the relationship.

Our second hypothesis was partially confirmed as well. Both parenting factors (authoritarian parenting style and the mother’s perception of the relationship as conflictual) were positively associated with the child’s problem behavior in school and this association was mediated by the child’s social cognition. Like in the previous case, however, these mediations were not held after background factors were entered into the equation. The links we found were associated with less positive maternal factors: Authoritarian parenting style and higher perception of conflict were linked to the child’s biased social cognition (lower levels of competent response generation), which, in turn, were associated with more problem behavior in school. Collectively, these findings support the limited data from previous research about the associations between negative parenting factors and children’s negative social cognition and maladjusted social functioning [22,23,32,33,34,36,39].

### 4.2. Clinical Implications

It is well established that children’s social adjustment early in life significantly predicts their social functioning in later stages of development [65]. Accordingly, there are numerous early intervention programs aiming to reduce children’s problem behavior and increase their prosocial behaviors as early as possible. Our findings could make an important contribution to such programs because they identify specific paths by which early maladjusted behaviors develop. In particular, the findings that social maladjustments are associated with biased and incompetent social cognition should be highlighted in that respect. Such findings support efforts to prevent children’s maladaptive behavior through targeting their social perceptions and, on the other hand, raise questions about the utility of behaviorally based interventions aiming to alter children’s social behaviors exclusively through means of instrumental reinforcements. Further, as our findings identify specific social information mechanisms associated with maladjusted social behaviors (i.e., response construction and generation), intervention programs aimed at social cognition could highlight these specific mechanisms in their efforts. Notably, such intervention programs are more common with elementary school-age children (e.g., Conduct Problems Prevention Research Group, [66,67]) but are almost non-existent in preschool. 

There are specific ways in which response generation could be targeted in intervention programs with children suffering from social difficulties. One successful and proven way is to create role-play activities in which children play different social roles in a controlled environment and under the close supervision of the teacher. In such situations, children could be directed to create different responses, and different outcomes to these responses could also be role-played. Teachers then provide feedback, correcting responses that are likely to bring social difficulties and encouraging social decision-making processes that are based on common social knowledge appropriate for a specific context. Programs such as Making Choices: Social Problem-Solving Skills for Children (MC; [68]) are based on intervention goals that are driven by social cognition, for example, pre-identify relational goals and design and select prosocial goals. 

Finally, our finding that the association between parent and child’s negative social cognition is mediated by authoritarian parenting style are also extremely important for clinical efforts. Such findings highlight the role of the parent in his/her child biased social perception. Clinically, it means that although programs such as those presented above could reduce the occurrence of problem behaviors, they are also limited because they do not target the likely main source of social cognitive distortions, the parent. Whereas such conclusions are problematic to our field as most intervention programs occur in the educational environment without the participation of parents, they should always be in the back of the mind of clinicians who truly want to change the development trajectories of children suffering from social problems.

### 4.3. Study Limitations and Future Directions

Even though this study includes many methodological advantages, some methodological limitations should be noted. First, data were collected at only one time point. Therefore, we cannot, at this point, empirically confirm a theoretically justified causal model that maternal general SIP predicts parenting factors that, in turn, predict the child’s SIP. It is entirely possible that these links occur in reverse order, especially, as noted before, in the association between authoritarian parenting style and the perception of the relationship with the child as conflictual. Additionally, we cannot confirm a causal link between parenting factors to child’s SIP, which, in turn, predicts maladaptive behavior. Alternative models in which, for example, the child’s status in school affects the relationship with the parent and her parenting style are also possible. To account for this limitation, future research may employ a longitudinal design in which, for example, the parent–child behavior would be assessed at the earliest point, child’s SIP would be evaluated at a middle point, and the child’s behavior would be measured at the final point. We are in the process of designing such a study in which the families participating in the current study will be followed into adolescence. Second, we did not include in our study measures assessing possible mediators/moderators that could shed further light on our findings and even change interpretations. Measures of parental psychological characteristics, as well as measures assessing parents’ knowledge of child development, are very important to more fully understand differences in parenting behaviors and perceptions. On the child’s side, assessing other psychological characteristics such as self-perception and motivation, as well as other social cognitive (e.g., the theory of mind) and cognitive (e.g., executive functions) factors, are crucial for a fuller understanding on the links between social cognition and social functioning. Adjusting for these limitations in future research may further advance our understanding of the possible links between parental perceptions and behaviors and children’s perceptions and behaviors. Finally, as this study is built on a moderate-size convenience sample, caution is advised in generalizing its findings.

## 5. Conclusions

As multiple developmental theories suggest, the link between the quality of the parent–child relationship and the child’s social behavior is quite solid. Our findings contribute to the line of research examining this link by providing evidence that this link not only exists, but it is indeed partially mediated by the child’s social information processing patterns. Additionally, for the first time, we were able to confirm an association between the parent and child’s SIP via parenting factors. The implications of these findings are important from both a theoretical and a clinical perspective. Theoretically, the study exemplifies the utility of a social information processing/social cognition approach to establish a more complete and succinct understanding of the links between a parent’s perceptions (general and child concrete) and behaviors toward her child and child’s behavior in a different social setting. Clinically, our findings are supportive of intervention approaches seeking to alter children’s thought processes as an effective means to alter their social functioning.

## Figures and Tables

**Figure 1 ijerph-17-00358-f001:**
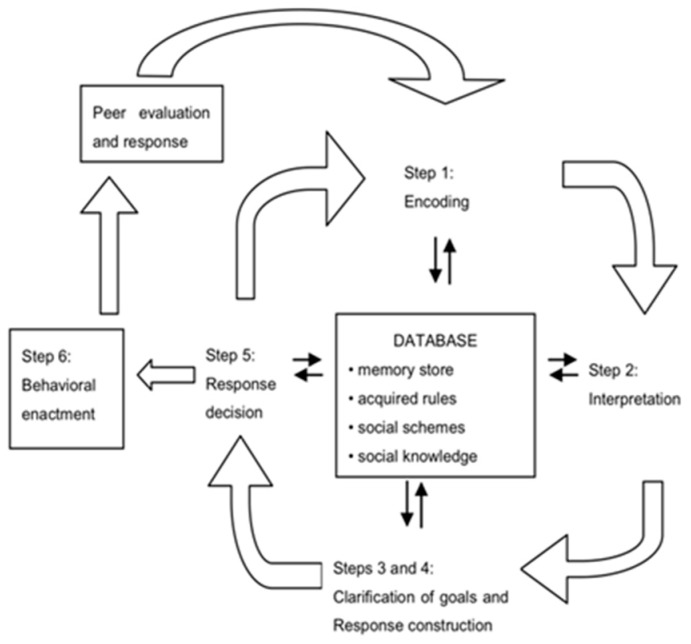
The social information processing model (based on Crick and Dodge, 1994).

**Figure 2 ijerph-17-00358-f002:**
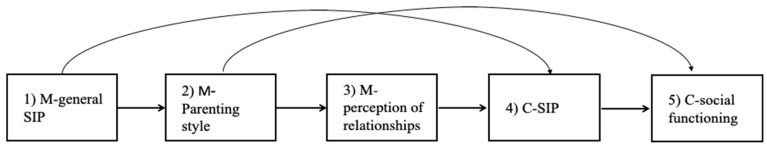
The study’s conceptual model.

**Figure 3 ijerph-17-00358-f003:**
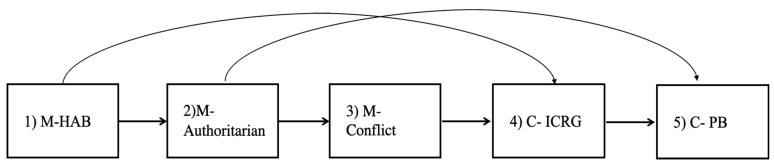
Study’s operational model and expected effects.

**Figure 4 ijerph-17-00358-f004:**
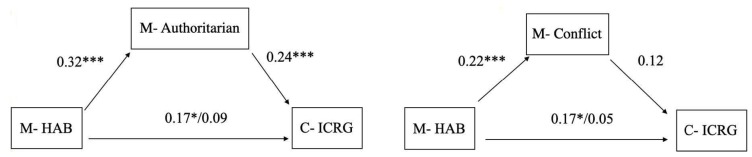
Mother’s authoritarian parenting style (**a**) and mother’s perception of conflict (**b**) mediate associations between mother’s hostile attribution and child’s competence generation response. Standardized coefficients are presented. Indirect effect for mother’s authoritarian parenting style, B = 0.08, SE = 0.02, *p* = 0.002, for mother’s perception of conflict, B = 0.02, SE = 0.02, *p* = 0.09. Mediation model for mother’s perception of conflict is adjusted for mother’s education and ethnicity. * *p* < 0.05; *** *p* < 0.001.

**Table 1 ijerph-17-00358-t001:** Descriptive statistics and correlations for person-level averages of all study variables and covariates.

Variable	1	2	3	4	5	6	7	8	9	10
1. MHAB	−									
2. Author	0.32 ***	−								
3. Conflict	0.18 *	0.57 ***	−							
4. C-ICRG	0.17 *	0.27 **	0.18 **	−						
5. C-PB	0.02	0.21 ***	0.20 **	0.17 **	−					
6. Child age	0.20 **	0.05	0.001	−0.09	0.03	−				
7. Income	−0.16 *	−0.23 **	−0.25 *	−0.12	−0.07	0.11	−			
8. MEDUC	−0.23 **	−0.31 ***	−0.34 ***	−0.25 **	−0.29 **	−0.05	0.35 ***	−		
9. Sector	−0.45 ***	−0.42 ***	−0.26 **	−0.23 ***	−0.07	−0.29 ***	0.12	0.31 ***	−	
10. Child’ sex	0.03	−0.10	0.02	0.02	0.17 **	−0.03	0.09	0.08	0.06	−
	M (SD)	M (SD)	M (SD)	M (SD)	M (SD)	M (SD)	M (SD)	M (SD)	%	%
	1.82 (1.68)	1.17 (0.48)	2.90 (0.47)	2.64 (1.34)	6.12 (5.19)	5.72 0.53	3.54 1.27	4.64 1.37	61.5% (Jews)	50.5% (girls)

MHAB = mother’s hostile attribution bias, Author = authoritarian parenting style, Conflict = mother’s perception of the relationship with the child as conflictual, C-CRG = child-incompetent response generation, C-PB = child-problem behavior. In all scores, higher scores represent more negative attributions/behaviors. Sex (girls = 0, boys = 1). Sector (Arabs = 0, Jews = 1). * *p* < 0.05. ** *p* < 0.01. *** *p* < 0.001.

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
