# Peer review of "Association between the Mother’s Social Cognition and the Child’s Social Functioning in Kindergarten: The Mediating Role of the Child’s Social Cognition"

_ijerph, 2020, doi:10.3390/ijerph17010358_

Round 1

Reviewer 1 Report

The manuscript entitled “Association between the mother's social cognition and the child's social functioning in kindergarten: the mediating role of the child's social cognition” is considered relevant for both the clinical and the theoretical fields. The aim of the study was to examine predictors of children's social functioning in kindergarten with a specific focus on maternal factors and children's social cognition.

In order to improve some of the sections of the manuscript, some comments are made that could be useful:

Introduction

My perception is that the introduction needs summarizing. The excessive detail in the presentation of ideas that are not relevant complicates the reading. I would recommend summarizing the ideas and presenting them more clearly to make the reading easier and friendlier.

Some ideas in the introduction are not supported by previous research. It is recommended that you include some references.

The name of the sections in the introduction should be changed to make them sound more attractive.

Materials and Method

The participants section needs to be separated from the procedure section. Moreover, the statistical analysis section should be in the method and not in the results.

The sections of the method should be as follows:

2.1. Participants

2.2. Instruments

2.3. Procedure

2.4. Statistical analyses

In the Participants section, it is required to put the SD of the age. Also, it would be advisable to indicate the mean and standard deviation of the mother’s age. Could the authors include the age range of adolescents and mothers?

In the section on Participants, the text refers to Table 1 to have more information on the participants, but this table contains other kind of information. There is a lack of results that give more information about the socio-demographic characteristics of the sample, such as parental level of education, familiar structure, members in the household...

It is recommended to include the time it took the adolescents to complete the questionnaires in the Procedure section.

Is it an incidental sample or was a probability sample used for the selection of participants?

Are the questionnaires validated for Israel's adolescent population? We recommend that the authors include the reference of each instrument.

Results

It is recommended to include the Sample Description in the Method section (Participants sub-section) instead of in the Results section.

I think it would be easier to understand the descriptive part if there was a table with the means and the typical deviations rather than just describing them in the text.

It is recommended not to include a section for each hypothesis. It would be more appropriate to integrate it in the text.

In figure 4, it is recommended to include, in the note, the meaning of the asterisks appearing in the figure.

Discussion

I would not separate the discussion into sections.

The study included only 301 children; therefore, caution is advised in generalizing the findings.

If probability sampling was not carried out for the selection of participants, it should be indicated as a limitation.

It is recommended to include throughout the discussion the conclusions of previous studies in order to understand whether or not they are in line with the results of the present study.

References

Authors are encouraged to review the reference format in depth. There are references in which the journal is not abbreviated, the “volume” appears, bold is missing, etc. It is suggested to follow the guidelines provided by the journal in the Guide for authors.

Author Response

Reviewer 1:

We want to thank you for the evaluation of our manuscript and appreciate the constructive nature of the review. We created a new version in which your queries were addressed.

For clarity, we address each of the questions and concerns on an item-by-item basis. (reviewer queries in bold font, our response in regular font).

My perception is that the introduction needs summarizing. The excessive detail in the presentation of ideas that are not relevant complicates the reading. I would recommend summarizing the ideas and presenting them more clearly to make the reading easier and friendlier.

Thank you for this suggestion. The introduction was reorganized and some sections were trimmed such that the main ideas are presented more clearly.

Some ideas in the introduction are not supported by previous research. It is recommended that you include some references.

We added some references to the introduction section wherever major ideas were not supported by previous research.

The name of the sections in the introduction should be changed to make them sound more attractive.

Thank you for this suggestion. However, the sections were named and organized according to our theoretical model and we believe they summarize the ideas within them quite well. 

A. The participants section needs to be separated from the procedure section. B. Moreover, the statistical analysis section should be in the method and not in the results.

We made these changes as requested. Participants section is now 2.1, procedure section 2.3, and analytic plan section, now in the Method, is section 2.4.

In the Participants section, it is required to put the SD of the age. Also, it would be advisable to indicate the mean and standard deviation of the mother’s age. Could the authors include the age range of adolescents and mothers?

We added mean age and SD to this section. We did not collect information about the mother's age. Additionally, the text in section 3.1 that elaborates on these variables was moved to the participants sub-section.

In the section on Participants, the text refers to Table 1 to have more information on the participants, but this table contains other kind of information. There is a lack of results that give more information about the socio-demographic characteristics of the sample, such as parental level of education, familiar structure, members in the household...

This information was already included in the Table. The bottom part include all the information requested about child's age and gender, sector, maternal education, and household income. As mention in the previous response, the full information is now included also in the participants sub-section.

It is recommended to include the time it took the adolescents to complete the questionnaires in the Procedure section.

This study did not include adolescents but preschool children. The interview time with the children was about 20 minutes as already written in section 2.2.2.

Is it an incidental sample or was a probability sample used for the selection of participants?

Although some efforts for a probability sampling were made, the sample should be characterized as a convenience sample.

Are the questionnaires validated for Israel's adolescent population? We recommend that the authors include the reference of each instrument.

Again, this is not an adolescent study. The interview used with the children was used extensively before in Israel. Additionally, all questionnaires used with parents and teachers were validated in previous studies in Israel. References for this previous work were already included in the manuscript.

It is recommended to include the Sample Description in the Method section (Participants sub-section) instead of in the Results section.

We moved this information to the participants sub-section.

I think it would be easier to understand the descriptive part if there was a table with the means and the typical deviations rather than just describing them in the text.

The information requested is included in the bottom part of Table 1.

It is recommended not to include a section for each hypothesis. It would be more appropriate to integrate it in the text.

Thank you for this suggestion but we respectfully disagree. We find this division helpful to the flow of the paper as well as for better understanding of the results.

In figure 4, it is recommended to include, in the note, the meaning of the asterisks appearing in the figure.

Added as requested.

I would not separate the discussion into sections.

We respectfully disagree. We find the division into separate sections helpful and friendlier to readers.

A. The study included only 301 children; therefore, caution is advised in generalizing the findings. B. If probability sampling was not carried out for the selection of participants, it should be indicated as a limitation.

We included the following sentence to the limitation part: "Finally, as this study is built on a moderate-size convenience sample, caution is advised in generalizing its findings."

It is recommended to include throughout the discussion the conclusions of previous studies in order to understand whether or not they are in line with the results of the present study.

This was done whenever appropriate (see for example, line 473)

Authors are encouraged to review the reference format in depth. There are references in which the journal is not abbreviated, the “volume” appears, bold is missing, etc. It is suggested to follow the guidelines provided by the journal in the Guide for authors.

We made all necessary changes as requested.

Reviewer 2 Report

Dear Authors,

I think your article is very interesting and have a high potential interest on this issue.

I have only one main concern with your research. I don't understand why do you have investigate the mother's knowledge of infant development and her social functioning.

It is likely that the association between mother and child's general social cognition will be mediated by the mother's knowledge of child development, which in turn affect parenting style. why did you not consider this variable? Please discuss you portion and intention also  examining some literature. Consider that mothers from different cultural backgrounds living together (in the same territory) might act differently and impact the adaptive behaviour of their children (Taverna, L., Tremolada, M., Bonichini, S. (2017). Conoscenze materne e sviluppo del bambino in due gruppi culturali altoatesini. Ricerche di Psicologia, 40(2), 257-278. DOI: 10.3280/RIP2017-002005). 

Another important thing concerns the Table 1. In the section 2.1 authors promise to give more details on participants' demographic characterstics which are not reported in Table 1 (correlations among variables and Mean and standard deviations).
Please provide descriptive information of participants.
61% are jewes and others only Arabs? How are children recruited in the study distributed with respect of age? Are there differnces among them?

Some minor things:

Line 321: scores = more
Line: portrayed is repetead two times in the same sentence
Table 1: columns are not justified, rows have not the same inter-space
Note: after author cancel the dot

Best regards and wish you success with your paper.

Author Response

Reviewer 2:

We want to thank you for the evaluation of our manuscript and appreciate the constructive nature of the review. We created a new version in which your queries were addressed.

For clarity, we address each of the questions and concerns on an item-by-item basis. (reviewer queries in bold font, our response in regular font).

I don't understand why do you have investigate the mother's knowledge of infant development and her social functioning. It is likely that the association between mother and child's general social cognition will be mediated by the mother's knowledge of child development, which in turn affect parenting style. why did you not consider this variable? Please discuss you portion and intention also  examining some literature. Consider that mothers from different cultural backgrounds living together (in the same territory) might act differently and impact the adaptive behaviour of their children (Taverna, L., Tremolada, M., Bonichini, S. (2017). Conoscenze materne e sviluppo del bambino in due gruppi culturali altoatesini. Ricerche di Psicologia, 40(2), 257-278. DOI: 10.3280/RIP2017-002005). 

Thank you for this important suggestion. In the limitation section, we listed a number of possible mediators and moderators that should be included in future studies if we wish to more fully understand the associations between parent and child's social cognition. As suggested, we added to this list 'knowledge of child development': "Measures of parental psychological characteristics as well as measures assessing parents' knowledge of child development are very important to more fully understand differences in parenting behaviors and perceptions."

Another important thing concerns the Table 1. In the section 2.1 authors promise to give more details on participants' demographic characterstics which are not reported in Table 1 (correlations among variables and Mean and standard deviations).
Please provide descriptive information of participants.
61% are jewes and others only Arabs? How are children recruited in the study distributed with respect of age? Are there differnces among them?

We think the reviewer may have missed the bottom portion of Table 1 in which all demographic characteristics are listed (columns 6-10). Additionally, more information is included in the participants sub-section (this information was moved there at the request of reviewer 1, see our responses to queries 5-6 above). Additionally, all the information about sector (Jews/Arabs) is also included and the differences between the groups were indeed tested and sector was included in the main analysis whenever such changes were found. The differences between sector groups are reported in Table 1, line 9, as correlations. Whenever required, sector was entered into the main analysis as covariate.

Some minor things:

Line 321: scores = more

We change the equation sign to the word "equals" so the meaning will be clearer

Line: portrayed is repetead two times in the same sentence

The second "portrayed" was changed to "shown"

Table 1: columns are not justified, rows have not the same inter-space

We do not understand this point. As we said before, we think the reviewer might have missed the second part of the table as it spread over two pages.

Note: after author cancel the dot

Dot should be there as this is an abbreviation of Authoritarian.